# Prognostic Relationship Between Human Papillomavirus Status and Overall Survival in Patients with Tongue Cancer

**DOI:** 10.3390/v17060780

**Published:** 2025-05-29

**Authors:** Chayanit Kritpracha, Peesit Leelasawatsuk, Virat Kirtsreesakul, Pasawat Supanimitjaroenporn, Jarukit Tantipisit, Manupol Tangthongkum

**Affiliations:** 1Department of Otolaryngology Head and Neck Surgery, Faculty of Medicine, Prince of Songkla University, Hat Yai 90110, Songkhla, Thailand; ckk.chayanit@gmail.com (C.K.); kvirat2002@hotmail.com (V.K.); ohm_ps1@hotmail.com (P.S.); 2Department of Pathology, Faculty of Medicine, Prince of Songkla University, Hat Yai 90110, Songkhla, Thailand; medew.jarukit@gmail.com

**Keywords:** human papillomavirus, prevalence, recurrence, survival, tongue cancer

## Abstract

An increasing number of patients are being diagnosed with tongue cancer despite lacking traditional risk factors such as tobacco, alcohol, or betel nut use. The potential role of human papillomavirus (HPV) in these cases has drawn attention, although its prognostic significance remains unclear. This retrospective cohort study, conducted in Southern Thailand, aimed to investigate the association between HPV status and overall survival in tongue cancer. Medical records of 186 patients treated between 2012 and 2021 were reviewed. HPV status was determined, and patients were categorized into HPV-negative, HPV-positive with other risk factors, and HPV-positive without known behavioral risk factors. Survival outcomes were compared using the log-rank test, and independent prognostic factors were analyzed using multivariable Cox regression. The overall prevalence of HPV infection was 9.6%, predominantly HPV16, HPV33, and HPV18. Although no significant differences were observed in 3-year overall survival, disease-specific survival, or recurrence-free survival among the groups, multivariable analysis revealed that HPV-positive patients without known behavioral risk factors (tobacco, alcohol, or betel nut use) had poorer overall survival (hazard ratio 3.54, *p* = 0.045). The observed associations warrant further research into the prognostic role of HPV in tongue cancer among non-smoking, non-drinking populations.

## 1. Introduction

Oral cancers rank among the top 10 most prevalent cancers worldwide [1]. These cancers are particularly prevalent in Asia, with over half of all cases occurring in this region, while 11% of cases originate from South East Asia [2]. The significant incidence of oral cancer in this region stems from the common habits of tobacco use, alcohol consumption, and betel nut chewing, which are established risk factors and poor prognostic factors for oral cancer. Despite improvements in universal healthcare and early diagnosis in Thailand in recent decades, the survival rates of oral cancer patients have not improved [3,4], emphasizing the importance of understanding the risk factors and maximizing treatment efficacy.

In recent years, an increasing number of patients have presented with oral cancer despite not having the typical risk factors. Globally, the tongue is the most commonly affected subsite of oral cavity cancers, accounting for approximately 25–40% of cases. Its incidence has been rising in both developed and developing countries [5]. This trend is particularly notable in the United States and parts of Europe, where an increasing number of younger individuals without traditional risk factors, such as tobacco or alcohol use, have been diagnosed with tongue cancer [6,7]. Among these cases, the tongue not only represents the most frequently affected site [6,8], but is also recognized as the most aggressive form of oral cancer. Recent data have brought attention to the potential role of human papillomavirus (HPV) in oral cancers, with strong correlations between HPV and oropharyngeal cancer, which has influenced changes in the staging system and potential implementation of treatment de-escalation [9]. The role of HPV infections in terms of causation and prognostic determination in oral cancers remains unclear. A recent study showed a significant prevalence of HPV infections in patients with oral cancer [10], suggesting a possible causative relationship between HPV infection and oral cancer. One meta-analysis found no clear correlation between HPV infection and overall survival (OS) in oral cancer patients [11]. However, several other studies found that HPV infection was favorable in terms of prognosis [10,11], while some others found unfavorable prognosis [12,13] and other studies found no prognostic correlations [14,15,16,17]. The heterogeneity of the data, resulting from the distinct characteristics of the different subsites, could be the source of these discrepancies [18].

The tongue is a major subsite of the oral cavity affected by cancer; to date, only a few studies have explored the possible association of HPV with tongue cancer. Several techniques have been used for HPV detection and genotyping, including polymerase chain reaction (PCR), E6/E7 HPV mRNA reverse transcription PCR, HPV DNA in situ hybridization, and p16 immunohistochemistry. The detection of p16 is ideal for oropharyngeal cancers due to its high sensitivity, specificity, and low cost, whereas in non-oropharyngeal cancers, p16 is weakly correlated with the gold standard and is not recommended for HPV testing [10,17,18,19]. Therefore, most studies on non-oropharyngeal cancers employ PCR-based methods for HPV detection.

To date, only a few studies have specifically examined the tongue to assess the correlations between HPV and survival outcomes, and information on HPV genotypes in tongue cancer is limited. Moreover, few studies with a large sample size used HPV DNA PCR to assess HPV status in these patients. Thus, this study aimed to provide insight into the association between HPV-related tongue cancers and OS, specifically in patients without any pre-existing risk factors.

## 2. Materials and Methods

### 2.1. Patient Selection

In this retrospective cohort study, the data of patients diagnosed with oral tongue cancer between January 2012 and December 2021 were extracted from the Cancer Registry Database of the Head and Neck Surgery Division of the Department of Otolaryngology, Faculty of Medicine, Prince of Songkla University, a major tertiary care center in Southern Thailand. The inclusion criteria were a diagnosis of squamous cell carcinoma of the tongue (ICD-10 topology code: C02.0–C02.3) and a receipt of curative treatment. Patients with base-of-tongue cancer (oropharyngeal cancer, ICD-10 topology code: C01) or a history of prior surgery, radiotherapy, or chemotherapy were excluded.

### 2.2. Ethics Approval Statement

Ethical approval for the study was granted, and the requirement for informed consent was waived due to the retrospective design of the study and its de-identified data analysis, by the Institutional Review Board, Faculty of Medicine, Prince of Songkla University (approval number: REC 65-192-13-1). All data were anonymized and stored in a secure database. We ensured that the study protocols adhered to relevant guidelines and regulations, including the ethical principles for medical research involving human participants, as provided by the World Medical Association (Helsinki).

### 2.3. Data Collection

All patients included in the study were pathologically confirmed to have squamous cell carcinoma. The cancers were staged according to the American Joint Commission on Cancer Staging Manual at the time of diagnosis. Data on demographic characteristics, ECOG performance status score, risk factors, tumor differentiation, pretreatment staging, treatment modality, date of diagnosis, date and status of last contact, and HPV status were collected. Smoking was defined as the current or former use of any tobacco products equal to or exceeding 20 packs–year. Alcohol use was defined as the regular consumption of alcoholic beverages exceeding four drinks per week. HPV status was categorized as HPV-negative, HPV-positive with other known risk factors (including smoking, alcohol, and/or betel nut use), or HPV-positive-only (defined as HPV-positive patients without the aforementioned behavioral risk factors). Treatment modalities were classified as surgery-based (surgery for early-stage cancer and surgery with postoperative radiotherapy or surgery with postoperative chemoradiotherapy for advanced cancer) or radiation-based (radiotherapy for early-stage cancer and concurrent chemoradiotherapy for advanced-stage cancer). Follow-up data were collected until 30 June 2023. The date of last clinical visit and date of death were recorded for all patients through a review of hospital medical records and, when necessary, verified via telephone contact. HPV detection and the genotyping of tongue cancer tissues, obtained from formalin-fixed, paraffin-embedded specimens, were conducted using the Anyplex II HPV28 assay (Seegene, Seoul, Republic of Korea). This assay employs multiplex real-time PCR to identify 28 HPV genotypes, including both high-risk (16, 18, 26, 31, 33, 35, 39, 45, 51–53, 56, 58–59, 66, 68–69, 73, 82) and low-risk (6, 11, 40, 42–44, 54, 61, 70) HPV subtypes [20,21]. The sensitivity and specificity of these tests in head and neck cancers have been shown to be 89.2% and 85.7%, respectively [21]. Data were analyzed using the Seegene Viewer program (https://www.seegene.com/software/seegene_viewer) (accessed on 8 July 2023).

### 2.4. Statistical Analysis

Descriptive statistics were employed to summarize frequencies and percentages, as well as medians and interquartile ranges, where appropriate. Differences in baseline clinicopathological characteristics and treatment modalities among the HPV-negative, HPV-positive with other risk factors, and HPV-positive-only groups were assessed using Fisher’s exact test. Survival probabilities for the three groups were compared using the log-rank test. Exploratory analyses of all potential variables associated with OS, DSS, and recurrence-free survival were conducted using a univariable Cox proportional hazards model. Multivariable Cox regression models were developed independently for each survival endpoint (OS, DSS, RFS), incorporating variables significant at *p* < 0.2 in univariate analysis and those deemed clinically relevant based on the existing literature. Given the distinct patterns and rates of events across these outcomes, variations in the selected variables and model outputs were anticipated. To verify that the core assumptions of the Cox regression models were maintained, a proportional hazards test was performed. A *p*-value < 0.05 was considered to indicate statistical significance. All statistical analyses were conducted using R software (https://www.r-project.org/) (accessed on 18 December 2023).

## 3. Results

### 3.1. Patient and Disease Characteristics

This study included 186 patients with tongue cancer, of whom 73 (39.2%) were female and 113 (60.8%) were male. The mean age of patients at diagnosis was 55 years. More than half of the patients (120, 64.5%) had advanced-stage tongue cancer. The prevalence of HPV infection was 9.6%, with HPV 16 being the most common subtype (14 cases, 7.5%), followed by HPV 33 (3 cases, 1.6%) and HPV 18 (1 case, 0.5%). Due to the low frequency of HPV33 and HPV18, subgroup analyses by individual HPV genotype, risk factor status, and survival outcomes were not statistically feasible. Consequently, all high-risk HPV-positive cases (HPV16, HPV33, and HPV18) were combined for the purposes of survival and risk factor analysis. The majority of patients (179, 96.2%) underwent surgical treatment. Considering the risk factors of smoking, alcohol consumption, and betel nut use, significant differences were found among the three groups: *p* = 0.004, *p* = 0.005, and *p* = 0.022, respectively (Table 1). The percentages of smokers, alcohol drinkers, and betel nut users were higher in the HPV-positive with other risks group than in the HPV-negative group. No significant differences in baseline clinicopathological characteristics and treatment modalities, including age group, sex, Eastern Cooperative Oncology Group (ECOG) score, underlying diseases (hypertension, diabetes mellitus, dyslipidemia, cardiovascular disease, pulmonary disease, and human immunodeficiency virus infection), differentiation, staging, and treatment, were found among the three groups.

### 3.2. Survival Outcomes

The median follow-up time was 27.9 (10.0–60.9) months. The 3-year OS rates of the HPV-negative, HPV-positive with other risk factors, and HPV-positive-only groups were 54.8%, 56.2%, and 25% (one of four patients), respectively. However, no statistically significant differences were found among the three groups (*p* = 0.506). Additionally, the 3-year disease-specific survival rates in the HPV-negative, HPV-positive with other risk factors, and HPV-positive-only groups were 57.8%, 68.2%, and 25% (one of four patients), respectively. Similarly to OS rates, no statistically significant differences were observed among the groups (*p* = 0.301) (Appendix A).

### 3.3. Disease Recurrence

The 3-year recurrence rates were 32.8%, 36.5%, and 50% in the HPV-negative, HPV-positive with other risk factors, and HPV-positive-only groups, respectively. The differences among the three groups were not statistically significant (*p* = 0.807) (Appendix A).

### 3.4. Multivariable Analyses of Survival

Cox proportional hazards models were used to conduct a multivariable analysis of OS (Table 2), which revealed that HPV-positive-only status (HR: 3.54, *p* = 0.045), smoking (hazard ratio [HR]: 2.02, *p* = 0.037), poor differentiation (HR: 2.63, *p* = 0.035), and tumor stage IV (HR: 3.44, *p* = 0.002) were independent variables associated with unfavorable survival outcomes. Similarly, HPV-positive-only status (HR: 4.19, *p* = 0.025), tumor stage IV (HR: 4.02, *p* = 0.002), and radiation-based treatment modality (HR: 2.91, *p* = 0.023) were independently related to unfavorable disease-specific survival outcomes (Table 3). Additionally, a multivariable analysis of recurrence-free survival (Table 4) showed that HPV-positive-only status had an HR of 1.8; however, the association was not statistically significant (*p* = 0.418).

## 4. Discussion

The prevalence of oral cancers in non-smokers and non-drinkers has increased in recent years [22], with the tongue being the most predominant subsite [23]. The clinical data suggest the aggressive nature of this disease entity. This study examined the association between HPV status and OS in a group of patients with tongue cancer, particularly focusing on individuals without other risk factors (tobacco use, alcohol consumption, and betel nut chewing). The prevalence of HPV infection in this study was 9.6%. Three distinct subtypes were identified: HPV 16, HPV 33, and HPV 18. Although the univariable analysis did not demonstrate any differences in OS, DSS, or recurrence rates, multivariable analysis found that HPV infection alone, without any additional risk factors, was a predictive factor for poor survival in patients with tongue cancer. Other independent factors associated with poor survival rates included smoking, poor tumor differentiation, radiation-based treatment modality, and stage IV disease.

Compared with a previous Taiwanese study [24] with similar patient demographics, our study found a relatively low prevalence of HPV-positive patients with tongue cancer (9.6% vs. 17.70%). The variations in prevalence could be due to differences in socioeconomic status and sexual behaviors among different regions [25,26]. Regarding survival outcomes, smoking risk, poor tumor differentiation, and stage IV cancer were found to be important prognostic factors for poor survival outcomes. However, patients with tongue cancer who were HPV-positive-only exhibited poor OS and disease-specific survival (DSS), with HRs of 3.54 and 4.19, respectively, after adjustment for confounding factors in multivariable analysis. This result was in agreement with the findings of Ramshankar et al. [27], which suggested that the presence of HPV in tongue cancers, in particular, was associated with a poor prognosis. Moreover, several large-scale studies by Nulton et al., Duray et al., and Lee et al. demonstrated a worse prognosis for patients with oral cancer when HPV infection was present [28,29,30], although subsite evaluations were not explored in detail. In addition, a more generalized study addressing head and neck squamous cell carcinoma reported that HPV infection, particularly HPV 33, had a strong negative impact on survival [31]. In contrast, some data have indicated that the presence of HPV is associated with improved OS in certain non-oropharyngeal squamous cell cancers [32]. Minami et al. [33] investigated tongue cancers specifically and reported that those with a p16-positive status had higher cause-specific survival. However, the clinical significance of this finding remains uncertain, as p16 is not a reliable surrogate marker for HPV infection in oral cancer [19,34,35], and confounding factors, such as smoking, were not included in the adjustment. Additionally, studies by Nauta et al. [36], Abreu et al. [37], and Schneider et al. [38] reported no association between HPV status and survival outcomes in patients with oral cavity squamous cell carcinoma. However, these studies did not specifically examine the oral tongue subsite. Another possible explanation for the lack of association in these studies is the confounding effect of smoking and alcohol use, which may have been mixed with HPV-positive status. In populations with a high incidence of smoking, these risk factors might impact the survival outcomes and neutralize the effect of the prognostic importance of HPV, as seen in our study and in others [39,40,41]. In our cohort, HPV-positive patients without known behavioral risk factors exhibited poorer survival outcomes compared to those with traditional risk factors. Although counterintuitive, this finding may reflect a distinct tumor biology in HPV-driven tongue cancers that occurs in the absence of other sources of carcinogenic exposure, or could be influenced by unidentified cofactors. These observations highlight the need for further molecular investigations to elucidate the underlying mechanisms. The reason for the correlation between HPV-positive status and poor prognosis is not well understood, although several possible explanations for this phenomenon could be postulated. First, HPV carcinogenesis is explained by p53 inactivation by the HPV E6 protein, which leads to cell cycle deregulation [35]. However, the molecular mechanism underlying the aggressive behavior and poor prognosis of HPV-positive tongue cancer remains unclear. Second, diminished DNA-mismatch repairs, resulting from the inactivation of proto-oncogenes, might account for the uncontrolled tumor growth [42]. Third, a molecular analysis of aggressive behavior in HPV tumors revealed an enriched signature of apolipoprotein B mRNA-editing enzyme, catalytic polypeptide-like (APOBEC), in HPV-infected head–neck cancers, causing DNA damage due to the APOBEC3A/B activity that occurs in response to viral infections [31]. Our current study found that HPV has promising benefits for predicting poor survival outcomes in patients with tongue cancer without any other risk factors, such as smoking, alcohol consumption, or betel nut chewing. However, while HPV-positive-only status was associated with poorer OS and DSS, no significant differences were observed in RFS. This may reflect the limited number of HPV-positive cases and events, resulting in reduced statistical power when detecting associations between these outcomes. In clinical settings, HPV testing can be easily conducted because of the accessibility of convenient and affordable assay kits. Furthermore, this test can be performed on existing pathology specimens obtained from diagnostic biopsies or surgical specimens, thereby eliminating the need for additional invasive procedures.

Our study included a large number of patients with tongue cancer. In addition, a multivariable analysis of survival outcomes showed good validity, as independent predictive factors influencing prognosis were identified. Moreover, this study provided insights into the HPV-positive-only group, using PCR as a testing method, which may represent a distinct entity of tongue cancer that arises from HPV-related causes. Given the relatively low prevalence of HPV-positive cases and the small sample size of the HPV-positive-only subgroup in our study, these findings should be interpreted with caution and considered exploratory. While our observations are consistent with some previous reports suggesting poorer outcomes in HPV-associated oral cancers, the current evidence specifically addressing the prognostic impact of HPV in tongue cancer remains limited. Therefore, our results should be regarded as hypothesis-generating and highlight the need for further validation in larger, prospective, multi-institutional studies focusing specifically on the tongue cancer subsite. The limitations of this study include its retrospective design and the relatively low prevalence of HPV-positive-only patients. The small number of cases for certain HPV genotypes limited our ability to perform a detailed analysis regarding the association between specific HPV types, risk factor exposure, and survival outcomes. Consequently, all high-risk HPV-positive cases were analyzed as a combined group. These limitations reflect the demographic characteristics of our study population. Future studies with larger sample sizes of HPV-positive-only patients, particularly in regions with higher HPV prevalence and lower smoking and alcohol consumption rates, along with longer follow-up periods, are needed to provide more comprehensive survival data and clarify the relationship between HPV status and tongue cancer.

## Figures and Tables

**Table 1 viruses-17-00780-t001:** Baseline clinicopathological characteristics and treatment modalities of study patients.

Variable	HPV-Negative	HPV-Positive with Other Risk Factors	HPV-Positive-Only	*p* Value
	(n = 168)	(n = 14)	(n = 4)	
Age group (years)				0.058
20–50	55 (32.7)	2 (14.3)	3 (75)	
51–70	94 (56)	10 (71.4)	0 (0)	
>70	19 (11.3)	2 (14.3)	1 (25)	
Sex				0.152
Male	101 (60.1)	11 (78.6)	1 (25)	
Female	67 (39.9)	3 (21.4)	3 (75)	
ECOG score				0.371
0	45 (26.8)	4 (28.6)	3 (75)	
1	118 (70.2)	10 (71.4)	1 (25)	
2	4 (2.4)	0 (0)	0 (0)	
3	1 (0.6)	0 (0)	0 (0)	
Underlying disease				
Hypertension	38 (22.6)	5 (35.7)	1 (25)	0.488
Diabetes mellitus	19 (11.3)	3 (21.4)	0 (0)	0.519
Dyslipidemia	19 (11.3)	3 (21.4)	0 (0)	0.615
Cardiovascular	4 (2.4)	1 (7.1)	0 (0)	0.402
Pulmonary disease	5 (3)	0 (0)	0 (0)	1
HIV	4 (2.4)	0 (0)	0 (0)	1
Smoking				0.004 **^†^**
No	75 (44.6)	2 (14.3)	4 (100)	
Yes	93 (55.4)	12 (85.7)	0 (0)	
Alcohol consumption				0.005 **^†^**
No	97 (57.7)	3 (21.4)	4 (100)	
Yes	71 (42.3)	11 (78.6)	0 (0)	
Betel nut chewing				0.022 **^†^**
No	134 (79.8)	7 (50)	4 (100)	
Yes	34 (20.2)	7 (50)	0 (0)	
Differentiation				1
Well	120 (71.4)	11 (78.6)	3 (75)	
Moderate	38 (22.6)	3 (21.4)	1 (25)	
Poor	10 (6)	0 (0)	0 (0)	
Staging				0.631
Stage I	29 (17.3)	2 (14.3)	1 (25)	
Stage II	30 (17.9)	4 (28.6)	0 (0)	
Stage III	33 (19.6)	3 (21.4)	2 (50)	
Stage IV	76 (45.2)	5 (35.7)	1 (25)	
Treatment				0.516
Surgery-based	162 (96.4)	13 (92.9)	4 (100)	
Radiation-based	6 (3.6)	1 (7.1)	0 (0)	

Note: The most common HPV subtype was HPV16 (14 cases), followed by HPV33 (3 cases) and HPV18 (1 case). No p16 immunohistochemistry data were available in this cohort. Data are presented as counts (%). ^†^ *p* values < 0.05 in Fisher’s exact test. Abbreviations: HIV, human immunodeficiency virus; HPV, human papillomavirus; ECOG, Eastern Cooperative Oncology Group.

**Table 2 viruses-17-00780-t002:** Univariable and multivariable analyses of overall survival in study patients.

Variable	Univariable HR	*p* Value	Multivariable HR	*p* Value
	(95% CI)		(95% CI)	
HPV status				
Negative	1 (reference)		1 (reference)	
Positive with other risk factors	1.09 (0.53, 2.26)	0.814	1.00 (0.47, 2.16)	0.996
HPV-positive-only	1.75 (0.55, 5.56)	0.341	3.54 (1.03, 12.20)	0.045 *
Smoking				
No	1 (reference)		1 (reference)	
Yes	1.34 (0.87, 2.07)	0.182	2.02 (1.04, 3.91)	0.037 *
Differentiation				
Well	1 (reference)		1 (reference)	
Moderate	0.95 (0.57, 1.58)	0.833	0.81 (0.48, 1.38)	0.435
Poor	2.32 (1.00, 5.38)	0.049	2.63 (1.07, 6.44)	0.035 *
Staging				
Stage I	1 (reference)		1 (reference)	
Stage II	1.94 (0.81, 4.63)	0.135	2.02 (0.83, 4.92)	0.122
Stage III	2.07 (0.89, 4.80)	0.090	1.77 (0.76, 4.16)	0.188
Stage IV	3.56 (1.69, 7.53)	<0.001	3.44 (1.59, 7.45)	0.002 *

* *p* values < 0.05. Abbreviations: HR, hazard ratio; CI, confidence interval; HPV, human papillomavirus.

**Table 3 viruses-17-00780-t003:** Univariable and multivariable analyses of disease-specific survival in study patients.

Variable	Univariable HR	*p* Value	Multivariable HR	*p* Value
	(95% CI)		(95% CI)	
HPV status				
Negative	1 (reference)		1 (reference)	
Positive with other risk factors	0.93 (0.40, 2.13)	0.855	0.77 (0.31, 1.92)	0.580
HPV-positive-only	1.91 (0.60, 6.09)	0.272	4.19 (1.20, 14.67)	0.025 *
Betel nut chewing				
No	1 (reference)		1 (reference)	
Yes	0.57 (0.31, 1.04)	0.065	0.69 (0.37,1.28)	0.238
Staging				
Stage I	1 (reference)		1 (reference)	
Stage II	2.05 (0.76, 5.54)	0.157	2.09 (0.76, 5.78)	0.155
Stage III	2.25 (0.87, 5.87)	0.096	1.83 (0.69, 4.88)	0.226
Stage IV	4.48 (1.91, 10.49)	<0.001	4.02 (1.66, 9.74)	0.002 *
Treatment				
Surgery-based	1 (reference)		1 (reference)	
Radiation-based	4.29 (1.96, 9.38)	<0.001	2.91 (1.16, 7.34)	0.023 *

* *p* values < 0.05. Abbreviations: HR, hazard ratio; CI, confidence interval; HPV, human papillomavirus.

**Table 4 viruses-17-00780-t004:** Univariable and multivariable analyses of recurrence-free survival in study patients.

Variable	Univariable HR	*p* Value	Multivariable HR	*p* Value
	(95% CI)		(95% CI)	
HPV status				
Negative	1 (reference)		1 (reference)	
Positive with other risk factors	1.19 (0.51, 2.76)	0.689	1.07 (0.46, 2.49)	0.881
HPV-positive-only	1.46 (0.36, 6.00)	0.597	1.8 (0.43, 7.52)	0.418
Sex				
Male	1 (reference)		1 (reference)	
Female	0.66 (0.39, 1.13)	0.129	0.66 (0.38, 1.13)	0.127
Treatment				
Surgery-based	1 (reference)		1 (reference)	
Radiation-based	3.26 (1.29, 8.20)	0.012	3.22 (1.27, 8.14)	0.013 *

* *p* values < 0.05. Abbreviations: HR, hazard ratio; CI, confidence interval; HPV, human papillomavirus.

## Data Availability

The data underlying this article will be shared on reasonable request to the corresponding author.

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
