# Peer review of "Prognostic Relationship Between Human Papillomavirus Status and Overall Survival in Patients with Tongue Cancer"

_viruses, 2025, doi:10.3390/v17060780_

Round 1

Reviewer 1 Report

Comments and Suggestions for Authors

This manuscript by Kritpracha et al. examines the prognostic relationship between HPV status and overall survival of patients with tongue cancer. This is an interesting study given that tongue cancer diagnosis in younger patients lacks known risk factors such as tobacco use, alcohol and betel nut chewing. Studies correlating the association between HPV infections and tongue cancer are lacking. In addition, there is a general lack of agreement among existing studies as to the role of HPV infections in progression of tongue cancer and overall survival. Here are my comments:

[1] In the introduction section, the authors talk about the epidemiology data relating to oral cancers in Asia/South East Asia. The authors should also include data from other parts of the world to demonstrate the importance of tongue cancer and association (or not) with HPV. The authors state that tongue cancer is not well studied and therefore it would be important to present a short summary of this problem as it stands globally.

[2] I was hoping to see a more detailed analysis of the high-risk HPV types that were determined, but they were presented as overall percentages in all the patients samples that were examined. The authors state the presence of HPV16, HPV18 and HPV33 in their samples but do not give details as to whether the presence of these types associated with the risk factors that were analyzed, for example tobacco, alcohol, betel nut chewing would be important. Could the authors please provide a more detailed description in this regard as well as overall survival.

[3] I am assuming that since the Anyplex II HPV28 kit is capable of detecting 28 different types of HPV including high- and low- risk types, the authors detected other HPV types apart from HPV16, 18 and 33 (?). Could the authors expand on their findings of the different HPV types detected and their association with the known risk factors and overall survival.

Reviewer 2 Report

Comments and Suggestions for Authors

The manuscript presents data and conclusions from 186 cases of tongue cancer. The lacking robustness of the HPV data would however require much more targeted and more accurate presentation, if the low number of HPV cases allows to draw conclusions on HPV at all. The authors conclusion on HPV impact in general is preceded by similar conclusions in the literature.

The presented data indicate the following to an expert:

-The known behavioral risk factors have stronger impact on tongue cancer than HPV does.

-The authors’ main conclusion surprisingly means, once tongue cancer has manifested itself, not only have the known behavioral risk factors more favourable prognosis than HPV does, but they also compensate the poor prognosis by HPV i.e. unbelievably authors insist on the HPV effect only in the absence of other risk factors.

Weak points of data and conclusion presentation questioning the surprising „HPV-positive-only” conclusion:

-The multivariate analysis is substantially different in relation to the three different outcomes suggesting that the authors might have performed a subjective modelling for multivariate analysis of each outcome to find results that are both positive and agreeing with those of previous publications.

-If HPV had a true impact, it should be seen for all three outcomes which are obviously closely realated. The authors made no attempt to explain why HPV could not act on disease-free survival.

-Although „HPV-positive-only” phrase can be used in most part of the manuscript, it is too slang in Abstract and in concluding remarks, the manuscript parts that would require precise definition of this patient group, if at all this patient group forms an entity different from the others.

Reviewer 3 Report

Comments and Suggestions for Authors

In the manuscript by Kritpracha et al. the authors describe a study of survival in tongue cancer patients depending on HPV status. While apparently robust on a first glance due to comprehensive medical data collection (ie comorbidities), clear design focus only on oral mobile tongue and suitable number of cases (n=186) assessed with adequate HPV detection methodology, the main focus of the study (survival analysis) is unusually obfuscated. Critically despite the original patients being treated at least 3 years ago (up to 13 years) the median follow up time was only approx. 2 years suggesting that most of the relevant data is actually missing. In line with this Kaplan meier plots were unusually absent.

While the above is only a descriptive note, the issues listed below (in page line format) warrant commenting or addressing:

P1 L30 given that authors rightly stress the regional differences in the introduction, it might be worthwhile to include the study location in abstract as well

P2 L77 ICD codes are shown as C02.0 - C02.3 on the ICD website.

P2 L78 it might be worthwhile to show ICD codes of excluded sites as well for consistency (C01 for base of tongue)

P3 L95 possibly the authors should describe what was considered smoking and or alcohol use for clarity

P3 L106 instead of reference 23 focusing on cervical cancer, authors should aim to include a reference supporting the use of Anyplex II HPV28 assay in FFPE tissues which can otherwise be challenging for PCR based methods testing for amplicons of large size.

P3 L113 given that the groups are so uneven it is good that Fischers exact test was applied. On the other hand, it is still hard to justify not combining the HPV positive groups together. Comparing 168 vs 14 vs 4 will certainly be very hard to interpret. Possibly the more relevant comparison would be 168 vs 18  while the 3 group comparison would be more appropriate for a supplementary table.

P3 L127 the HPV genotyping data should include numbers and not only percentages. Furthermore, this might warrant inclusion in Table 1 to clearly show where the non-HPV16 types were found. Was there p16 data available?

P4 L138 Table 1 should not be split across pages. Same applies to other tables (ie Table 3)

P5 Table 1. Maybe aligning the variable names (Sex, ecog score…) to the right would make the table rows more readable and sublevels more noticeable. Same applies for other tables (ie Table 2,3,4)

P4 L143. It is unusual that the median followup is only ~2 years given that patients were collected between 2012 (13 years ago) through 2021 (3 years ago). Since the followup data collection is not declared in the methods section this suggests that vast majority of patients were lost to followup. When was the followup data collection cutoff date, and what methods were employed to collect followup data?

P4 L145 the problem of a very small HPV positive group (n=4) becomes apparent again since the 3Y OS of only 25% (1 out of 4) is very strongly dependant on chance. It is critical that "(1 of 4)" is added to the text for clarification at this line and elsewhere as not to mislead readers in the robustness of this percentage shown.

P4 L142-149 Kaplan Meier graph of the survival curves should be shown to strengthen the results and possibly explain the problems of followup data. Possibly OS, DSS, and PFS plots can be shown as 3 panels. Or at least as supplementary figures

Furthermore, this section only shows OS and omits DSS which is later used in Cox regression (ie Table 3)

Reviewer 4 Report

Comments and Suggestions for Authors

This study is very good written and performed. Discussion part is particularly good written with adequate literature preview. I suggest accepting after minor corrections.

  1. Results

In this chapter you claim that: “The 3-year OS rates of the HPV-negative, HPV-positive with other risk factors, and HPV-positive-only groups were 54.8%, 56.2%, and 25%, respectively. However, no statistically significant differences were found among the three groups (P = 0.506).” … please, explain better (because of the small number of samples) …

  1. Conclusions

This chapter should be enlarged.

Round 2

Reviewer 1 Report

Comments and Suggestions for Authors

Thank you for the revised manuscript !

Author Response

Comment: Thank you for the revised manuscript!

Response: We sincerely appreciate your kind acknowledgement and thank you for taking the time to review our revised manuscript. Your feedback and support are greatly valued.

Reviewer 2 Report

Comments and Suggestions for Authors

After disapproving the previous manuscript version at all, these are my comments on the revised version:

The corrections are right but do not improve on scientific soundness. You cannot draw these conclusions from you data even if they agree with those of others.

I can approve the publication this manuscript only if the last sentence of Abstract is deleted and the 5. Conclusion part is deleted, too. 

Reviewer 3 Report

Comments and Suggestions for Authors

all comments were addressed and/or limitations acknowledged

Author Response

Comment: all comments were addressed and/or limitations acknowledged

Response: We sincerely appreciate your kind acknowledgement and thank you for taking the time to review our revised manuscript. Your feedback and support are greatly valued.